# Numerical Analysis of the Free-Falling Process of a Water Droplet at Different Temperatures

**Yuchao Song** [1,*] **, Yafei Zhang** [2] **and Hongtao Gao** [2]

1. Marine Engineering College, Dalian Maritime University, Dalian 116026, China
2. Naval Architecture and Ocean Engineering College, Dalian Maritime University, Dalian 116026, China
* Correspondence: dllmusc@dlmu.edu.cn

**Abstract:** The collision behavior and ice formation of a water droplet are affected by its falling process. In this paper, the two-phase flow of air and a water droplet at a specific temperature is adopted to investigate the processes of falling and freezing of a single water droplet. To track the air–water droplet interface and the temperature distribution, the level-set method and the non-isothermal flow coupling method are used, and the freezing model is added into the water's control equations. The numerical results indicated that with the initial temperature at 283.15 K and the spherical shape, the water droplet changes to the shape of a straw hat at 293.15 K and a drum at 293.15 K but an oval face in freezing temperatures at 0.10 s. There is an obvious drop in the downward velocity when the water droplet falls in mild temperatures at 0.09 s. The downward velocity of the water droplet in air at sub-zero temperatures has a continuous increase during the time span from 0 s to 0.10 s. There is also an obvious difference when the water droplet impinges on the solid bottom. Lastly, the freezing of sessile water droplets attached on the horizontal surface is helpful to reveal the unique phase change process of water droplets in cold air.

**Keywords:** two-phase flow; heat transfer and phase change; water droplet; deformation

## 1. Introduction

Water droplets are very common in nature and engineering applications. The dynamic freezing of water droplets in different environments has emerged as a scientific problem in recent years. As the attached surfaces, there are static structure surfaces, translational object surfaces and rotating blade surfaces, which involve various engineering fields such as sea, land and air. The freezing of water droplets brings a great challenge to scientists. Extensive research on this topic is still needed, and the demonstration of dynamic freezing can also provide a scientific basis for deicing and anti-icing [1].

The dynamic freezing process of water droplets can be divided into two main stages: free falling in front of the impinging cold wall and crystallization freezing after contact. It is generally believed that the shape of the falling water drops will deform from the beginning of landing, and the deformation order is spherical, ellipsoidal and spherical cap/ellipsoidal cap [2,3]. When a water droplet comes into contact with the wall at room temperature, the following will occur successively: adhesion, rebound, spreading and splashing [4]. When the surface temperature is lowered to a sub-zero temperature, it is similar to sessile water droplet freezing, as the freezing process of a water droplet landing on a cold surface is most affected by the wall temperature and ambient temperature [5]. However, only when the wall temperature further drops will the effect of impact velocity on icing be weakened [6]. Water droplets have different shapes, ice formation, vibration and noise during impact [7]. The impact velocity, wettability and deformation of water droplets at 25 °C were studied by Meng-Jiy Wang and et al. [8,9]. They found that the droplet spreading velocity increased with the increase in impact velocity (Weber number), and that the droplet spreading speed also increased [10]. At the same time, higher impact velocity and elliptical water droplets

can produce larger bubble entrainment behavior [11]. In addition to room temperature, it was found that the increase in the initial height of water droplets shortens the icing time [12], and the downward velocity of water droplets in cold air decreases [13]. There are two methods to study the interaction between water droplets and air. Zhang [14] and Huang [15] conducted freezing experiments, and the results showed that the freezing time grows with the increase in contact angle. During the falling and impinging test, when the plate surface temperature is relatively high, the droplets present a tip shape in the middle, and when the plate surface temperature is low, a new concave annular shape appears [16]. In an experiment and numerical investigation on the freezing characteristics of sessile water droplets submerged in silicone oil at sub-zero temperatures taken at the same time [17], it was found that the fluid is incompressible and Newtonian, its flow is two-dimensional and laminar, and the water droplet remains round all the time. The temperature difference and pressure gradient inside the droplet form a strong recirculation region inside the droplet, and the freezing develops from outside to inside. Different from the water droplet freezing on the horizontal plate, Jin [18] experimentally investigated the impact, freezing and melting processes of a water droplet on an inclined cold surface, and the water droplet became opaque immediately when it stayed on the cold wall.

In terms of numerical simulation, the macro computational fluid dynamics method is mainly used to simulate the phase change and icing process of static water droplets [19]. With a series of micro-structured arrays with squared cross-sectional pillars, the volume of fluid (VOF) method is used to simulate the complicated freezing process of micro-sized water droplets (diameter smaller than 100 μm) on the superhydrophobic surfaces [20]. With the assistance of the level-set method, the propagation of the ice front during the freezing process was conducted in a 3D numerical model [21]. The phase-field method is used to predict a concave shape of the freezing front, and a projection is formed at the last freezing stage [22]. The bubble formed during freezing is a factor to bring this projection [23]. The model was validated using STAR CCM+ software to study the collision of a partially frozen droplet with a solid wall [24]. The free-falling process of water droplets is important to study the impact, spreading and icing of water droplets. Sultana [25] numerically examined the phase change of free-falling droplets in a sub-zero environment, and the falling altitude was about 10 m; with the initial nucleation at the droplet top surface, the freezing grew rapidly inwards from the surface of the droplet. This falling process needs to be further revealed, especially when the water droplet falls in air at freezing temperatures.

In this paper, the falling, deformation and temperature changes of water droplets in different specific temperature environments are simulated and analyzed. The two-phase flow method was used to study the dynamic behavior of water droplets before collision. At the same time, considering the heat transfer between the frozen solid bottom and the fluids, the outside and inside characteristics of the water droplet are illustrated.

## 2. Force Description of a Falling Water Droplet

Water droplets falling in air are affected by the gravity $F_1$, the buoyancy $F_2$ and the drag $F_3$ [26]. These forces acting on a water droplet can be expressed by the following formulas:

$$M_{wd} \frac{\mathrm{d}u_{wd}}{\mathrm{d}t} = F_1 + F_2 + F_3 \tag{1}$$

$$F_1 = \rho_{wd} \, V_{wd} \, g \tag{2}$$

$$F_2 = \rho_a \, V_{wd} \, g \tag{3}$$

$$F_3 = -\frac{1}{2} \, \lambda_d \, A_{wd} \, \rho_a \, (u_{wd} - u_a)|u_{wd} - u_a| \tag{4}$$

where $M_{wd}$ is the mass of a water droplet (kg), $u_{wd}$ is the velocity of a falling water droplet (m/s), $u_a$ is the velocity of the air (m/s), $A_{wd}$ is the surface area of a water droplet in the direction of motion (m$^2$), $V_{wd}$ is the volume of a water droplet (m$^3$), $\rho_{wd}$ is water density

(kg/m$^3$), $\rho_a$ is air density (kg/m$^3$), $g$ is gravity acceleration and $\lambda_d$ is drag coefficient. The drag coefficient $\lambda_d$ is related to the Reynolds number $R_e$ [27], and its expression is

$$\lambda_d = \frac{24}{R_e}, \ R_e \ \leq \ 1 \tag{5a}$$

$$\lambda_d = \frac{24}{R_e}\left(1 + 0.14 R_e^{0.7}\right), \ 1 \leq \ R_e \ \leq \ 1000 \tag{5b}$$

and the Reynolds number $R_e$ is expressed as

$$R_e = \frac{\rho_a \ D_{wd} \ v_s}{\mu_a} \tag{6}$$

where $D_{wd}$ is the diameter of a water droplet, $v_s$ is the velocity difference between a water droplet and air and $\mu_a$ is the air's viscosity.

## 3. Two-Phase Flow Model of Water Droplet and Air

### 3.1. Numerical Model and Governing Equations

Considering the continuity, momentum and energy, as well as the influence of freezing temperature on the physical parameters of water droplets and air, the air is modeled with the default equations (the applicable temperature range of air parameters is from 200 K to 1600 K in COMSOL). For water droplets, the freezing control mode ($T \leq 273.15$ K), as presented by Equations (7)–(10), is added into the original model (273.15 K < $T$ < 373.15 K).

$$\mu_{wd} = 1 \tag{7}$$

$$C_{pw} = 7.178 \ T + 141 \tag{8}$$

$$\rho_{wd} = -0.163 \ T + 961.4 \tag{9}$$

$$k_w = -0.011 \ T + 5.22 \tag{10}$$

where $\mu_{wd}$ is the dynamic viscosity, $C_{pw}$ is the heat capacity, $\rho_{wd}$ is the density and $k_w$ is the coefficient of heat conductivity for water when 273.15 K < $T$ < 373.15 K.

The model of a single water droplet in an air tube is shown in Figure 1. The initial radius and height of the spherical water droplet are $R$ and $h$, the size of air pipe field is $B \times H$ and gravity $g$ is considered. Concretely, the geometry sizes are $R = 2$ mm, $h = 90$ mm, $B = 20$ mm and $H = 100$ mm. During the free-falling process, the water droplet is affected by air resistance $F_3$ and air temperature $Tg$. The heat transfer also occurs between the water droplet and its surrounding air. These characteristic parameters of the water droplet during the falling process, including deformation, velocity and temperature changes, are studied in the following analysis.

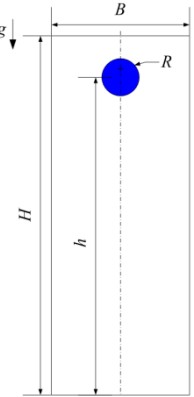

**Figure 1.** The geometric sketch of water droplet in an air pipe.

To describe the falling behavior of the water droplet, the fluid model used in the numerical calculation is a multi-physical field model set with the two-phase laminar flow, the level-set method and the non-isothermal flow. The surface tension force of water in air is 0.0072 N/m.

*3.2. Grid Study and Validation*

The deformation grid domain is used to simulate the deformation of water droplet. The meshing of the numerical model is based on the principle of fluid dynamics refinement, and the boundary layer grids are set at the boundary of the water droplet. Generally, the accuracy of the numerical computation depends on the resolution of the computational grid. Four cases of grids are shown in Table 1. For the water droplet domain, there is the best grid condition in case 3, considering the mean quality and the lowest quality together (1 is the highest grid quality). A computational case of the water droplet falling in air at 283.15 K is shown in Figure 2. The grid quality is close to 1, but the shapes of the water droplet are different from each other. Due to the coarser grid, there is a rough edge in case 1 at 0.05 s. In cases 2 and 3, the water droplet is like a human face shape, but in case 4, the shape is ellipsoidal. Therefore, the grid in case 3 is chosen to perform the falling simulation in this paper.

**Table 1.** Case studies for computational grid resolution.

| Case | Mean Quality of Grid | The Lowest Quality of Grid |
|---|---|---|
| Case 1 | 0.8540 | 0.7316 |
| Case 2 | 0.8368 | 0.6929 |
| Case 3 | 0.8740 | 0.7633 |
| Case 4 | 0.8768 | 0.6960 |

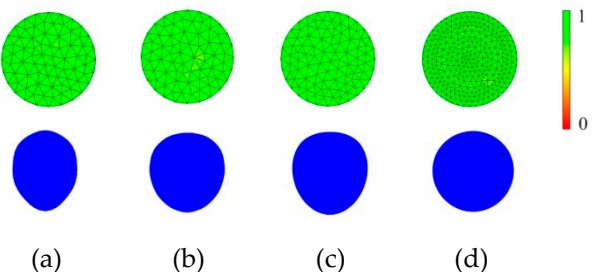

**Figure 2.** Grid distribution of water droplet and its shape falling in air at 283.15 K, 0.05 s: (**a**) case 1; (**b**) case 2; (**c**) case 3; (**d**) case 4.

## 4. Numerical Results of the Free-Falling Process

Falling in the air at a specific temperature, the initial drop height of water droplets remains the same, and the initial temperature is constant at 283.15 K. Considering that the air temperature is 293.15 K, 283.15 K, 263.15 K, 253.15 K and 243.15 K, respectively, there are five falling cases. The hydrodynamic characteristics of the falling processes were analyzed.

When the water droplet drops, it is forced to deform under the influence of the surface tension and air drag forces. The corresponding water droplet deformation $A$ has the following expression [28]:

$$A = \frac{C_F \rho_{wd} v_{wd}^2}{C_k} \cdot \frac{R^2}{\sigma} \tag{11}$$

where $\sigma$ is surface tension coefficient, $R$ is initial radius of water droplet and $C_k = 8$, $C_F = 0.33$.

The deformation of a water drop is the decisive factor for deformation and breakage of a water drop. Table 2 shows the deformation factor $A$ of a water droplet when the water droplet falls at different air temperatures at 0.02 s, 0.06 s, 0.08 s and 0.10 s. These values

calculated by Equation (11) indicate that the water droplet deforms during its falling at different times. The air temperature also affects the water droplet's deformation. When the air temperature is 283.15 K, the deformation *A* is a little larger than that in other air temperatures at 0.02 s, and the deformations of a water droplet in other air temperatures, such as 293.15 K, 263.15 K, 253.15 K and 243.15 K, are similar to each other. Until 0.09 s, the deformation factor *A* increases all the time, but it increases more markedly at mild temperatures. When the mild temperature is 293.15 K at 0.09 s, the deformation factor *A* increases to a high value of about 0.014; while in freezing air, the deformation factor *A* is just about 0.011. As the water droplet continues to fall, the droplet in the mild temperature has a smaller deformation factor before the water droplet impinges on the bottom. These values of deformation factor *A* indicate that the water droplet suffers a large deformation before impact. Analyzing all the deformation values, the deformation factor is far less than 1 in the falling process of the water droplet, and there is no fracture phenomenon but there is an obvious deformation.

**Table 2.** Deformation factor *A* of a falling water droplet ($10^{-4}$).

| Tg (K) | t = 0.02 s | t = 0.04 s | t = 0.06 s | t = 0.08 s | t = 0.09 s | t = 0.10 s |
|--------|-----------|-----------|-----------|-----------|-----------|-----------|
| 293.15 | 7.84 | 34.86 | 73.16 | 115.52 | 135.87 | 120.46 |
| 283.15 | 8.27 | 33.97 | 71.87 | 118.80 | 146.67 | 120.46 |
| 263.15 | 7.30 | 30.53 | 60.78 | 93.87 | 110.69 | 125.49 |
| 253.15 | 7.59 | 30.53 | 60.78 | 93.87 | 110.69 | 127.19 |
| 243.15 | 7.30 | 29.70 | 60.78 | 93.87 | 110.69 | 125.49 |

Figure 3 shows the specific deformation shape of the water droplet falling in the air at a specific temperature. During the falling process, the droplet changes from the initial sphere to other shapes. At 293.15 K, the water droplet deformation is in the order of first a sphere, then an ellipse, at last a straw-hat shape. At 283.15 K, the deformation process is similar to that at 293.15 K, but it is a shape like a drum at 0.10 s. The deformation in freezing air is obviously different from that in air with a mild temperature. For example, the water droplets elongated significantly in the vertical direction after 0.08 s in air at 263.15 K, resulting in a melon seed deformation. With the decrease in air temperature, the vertical elongation decreases, and the deformation shape is similar to an oval face.

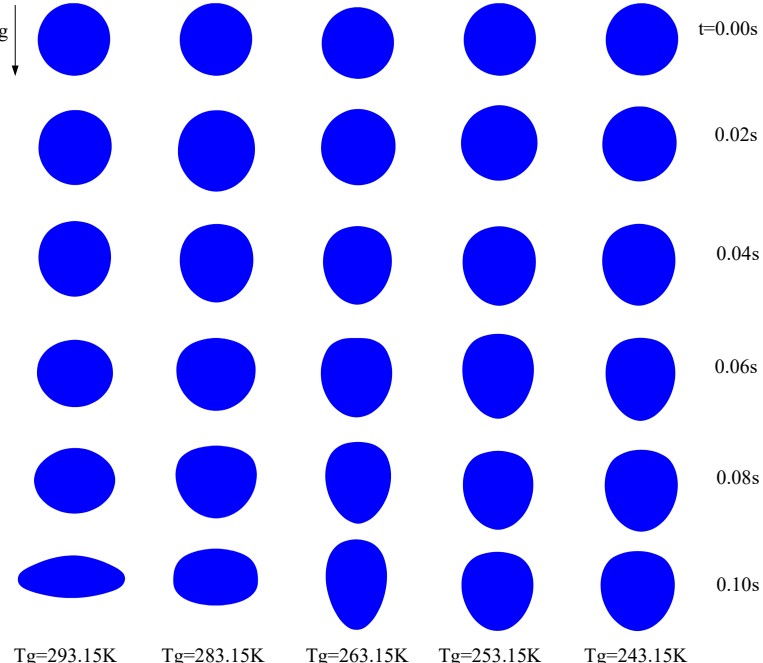

**Figure 3.** Deformation shape of free-falling water droplets at different temperatures.

The deformation ratio *E* is also used to describe the deformation degree of the water droplet. It is defined as the ratio between the length of the vertical axis and the length of the horizontal axis of one water droplet. The value of this ratio *E* represents the deformation degree of the droplet to a certain extent. For spherical droplets, *E* = 1. For a flat ellipsoid, *E* < 1. For a vertical ellipsoid, *E* > 1. Figure 4 is the curve of deformation ratio *E* values during the water droplet falling. As the dropping of water droplets takes place, the water droplet in air at a mild temperature is firstly enlarged in the vertical direction, and then enlarged in the horizontal direction. The water droplet in air at a freezing temperature has a persistent enlargement in the vertical direction. Regarding the deformation *E*, the largest value is about 1.5 in air at 263.15 K, and the smallest value is about 0.4 in air at 293.15 K.

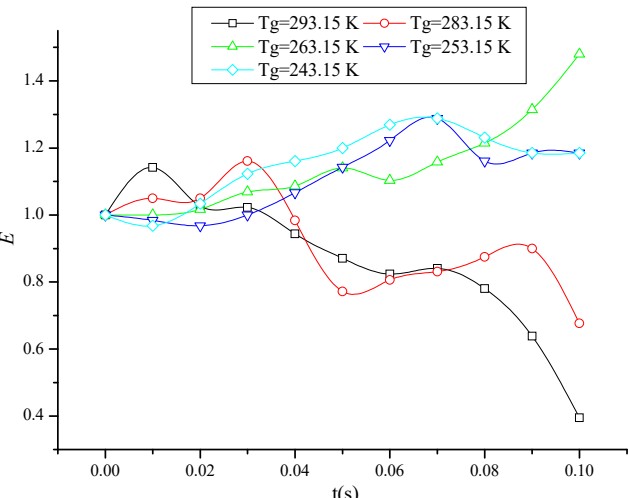

**Figure 4.** Variation in water drop deformation ratio *E*.

## 5. A Test of Free Falling

To reveal the characteristic process of the falling water droplet in an air pipe, the free falling of a single water droplet in air at 283.15 K was tested at the Institute of Refrigeration and Cryogenic Engineering in DMU. A water feeding device, a glass tube and a high-speed camera (Phantom V310) were used in this test. During the test, the sample rate of the high-speed camera was set at 6000 fps.

Figure 5 shows the free-falling process and the deformation of the water droplet at an air temperature of 283.15 K. In this test, the water droplet falls with a spherical shape, then it changes to an ellipsoidal shape, and at last it remains the ellipsoidal shape before the water droplet impinges on the bottom. The free-falling test and the numerical analysis at 283.15 K have a mostly similar phenomenon regarding the water droplet deformation. Due to the equipment conditions in the laboratory, other temperature tests will be conducted in the next test research. As there is reliable accuracy in this 2D numerical model, the flow analysis results can be accepted in this paper.

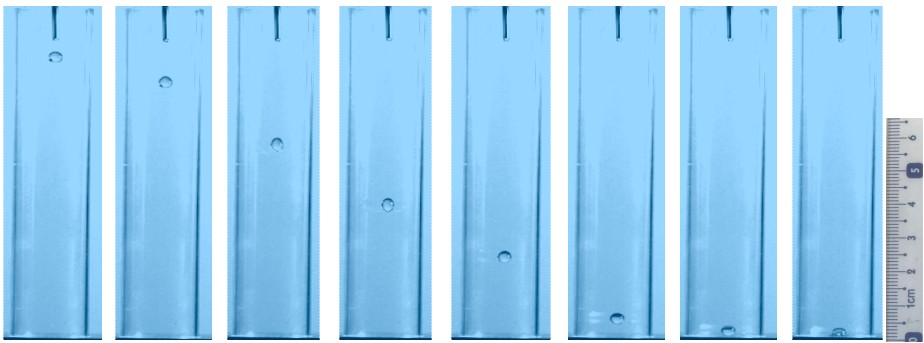

**Figure 5.** Deformation shape of free-falling water droplet in test at 283.15 K.

## 6. Discussion of Characteristic Parameters

### 6.1. Velocity of Free-Falling Water Droplet

The falling process of water droplets can be quantitatively described by Reynolds number, Weber number, Froude number and Ohnesorge number [29]. Reynolds number is the ratio of inertial force to viscous force, Weber number is generally used to measure the ratio of surface tension to inertial force, Froude number is used to describe the ratio of inertial force to gravity and Ohnesorge number is used to describe the ratio of viscous force to surface tension. These expressions of Froude number, Weber number and Ohnesorge number are as follows.

$$F_r = \frac{C_F \rho_{wd} v_{wd}^2}{C_k} \cdot \frac{R^2}{\sigma} \tag{12}$$

$$W_e = \frac{\rho_a d_w v_s^2}{\sigma} \tag{13}$$

$$Oh = \frac{\eta_s}{\sqrt{\rho_w \sigma d_w}} = \frac{\sqrt{W_e}}{R_e} \tag{14}$$

where the water droplet diameter $d_w$ is determined by the size of the major axis and minor axis of the water droplet [3].

From the initial position, the water droplet falls and suffers the forces described by Equation (1). Figure 6 shows the velocity curves of water droplets in the falling process. As the falling height is constant, the velocity of the water droplet increases linearly with time from 0 s to 0.08 s, and there is a similar trend during this time span. During the time span from 0 s to 0.03 s, all the velocity curves have the same trend. After 0.03 s, the water droplet in air at a mild temperature has a slightly higher velocity. When the temperatures are 293.15 K and 283.15 K, the highest velocity of the water droplet appears at 0.09 s, and the highest velocity is 1.54 m/s and 1.60 m/s, respectively. After 0.09 s, the water droplet's velocity at 293.15 K and 283.15 K has a fast drop, due to impinging on the bottom. When the freezing temperatures are 263.15 K, 253.15 K and 243.15 K, the velocity increases continuously from 0 s to 0.10 s. The linear trend gradient is about 14.89. The curve trends of water droplet velocity in these freezing temperatures are very close to each other, rather than that at mild temperature. Due the fast falling, the water droplet in the air a at mild temperature impacts on the bottom at 0.10 s, but the water droplet in air at a freezing temperature impacts the bottom just after 0.10 s. The maximum velocity values of this falling analysis are 1.60 m/s at 283.15 K at 0.09 s and 1.39 m/s at the freezing temperature at the same time. The velocity at the freezing temperature is 13.13% lower than the velocity of the water droplet at 283.15 K. From these velocity values, it can be seen that there is an approximate linear increase in downward velocity during all these free-falling processes, but the water droplet in the air at a mild temperature has a higher velocity than in air at a freezing temperature.

According to the definition of Reynolds number, Froude number, Weber number and Ohnesorge number in Equations (6) and (12)–(14), as well as considering the change in physical parameters with temperature, the values of these parameters at 0.06 s and 0.10 s are shown in Table 3. It can be seen from the equivalent diameter that the deformation of water droplets will cause the reduction in the equivalent diameter. All the Reynolds numbers are less than 550, and the free-falling process of water droplets belongs to two-phase laminar flow. However, the Reynolds number of water droplets in the freezing environment is a little higher than that in the mild temperature environment. In this paper, the Froude number is less than 10, the Weber number is less than 0.2 and the Ohnesorge number is less than 0.002.

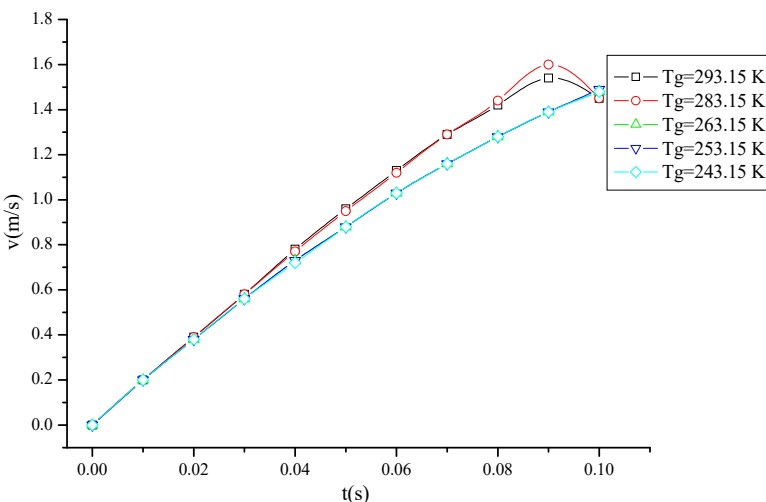

**Figure 6.** The velocity of the free-falling water droplet in air at different temperatures.

**Table 3.** Parameters of the free-falling water droplet.

| Time | Tg (K) | $V$ (m/s) | $D_w$ (mm) | $R_e$ | $F_r$ | $W_e(10^{-2})$ | $Oh(10^{-4})$ |
|---|---|---|---|---|---|---|---|
| 0.04 s | 293.15 | 0.78 | 3.11 | 161.56 | 4.47 | 3.17 | 11.02 |
| | 283.15 | 0.77 | 3.03 | 165.04 | 4.47 | 3.11 | 10.68 |
| | 263.15 | 0.73 | 3.03 | 177.60 | 4.24 | 3.01 | 9.76 |
| | 253.15 | 0.73 | 3.10 | 194.97 | 4.19 | 3.20 | 9.18 |
| | 243.15 | 0.72 | 3.03 | 202.13 | 4.16 | 3.17 | 8.81 |
| 0.09 s | 293.15 | 1.54 | 3.27 | 335.21 | 8.60 | 12.98 | 10.75 |
| | 283.15 | 1.60 | 3.00 | 340.68 | 9.32 | 13.32 | 10.71 |
| | 263.15 | 1.39 | 3.16 | 352.69 | 7.90 | 11.37 | 9.56 |
| | 253.15 | 1.39 | 2.96 | 354.36 | 8.16 | 11.08 | 9.39 |
| | 243.15 | 1.39 | 2.96 | 380.85 | 8.16 | 11.54 | 8.92 |
| 0.10 s | 293.15 | 1.45 | 4.27 | 412.15 | 7.09 | 15.02 | 9.40 |
| | 283.15 | 1.45 | 4.00 | 411.48 | 7.32 | 14.58 | 9.28 |
| | 263.15 | 1.48 | 4.16 | 494.46 | 7.33 | 16.97 | 8.33 |
| | 253.15 | 1.49 | 3.96 | 508.16 | 7.56 | 17.04 | 8.12 |
| | 243.15 | 1.48 | 3.96 | 542.48 | 7.51 | 17.51 | 7.71 |

### 6.2. Pressure Distribution of Water Droplet

When the water droplet drops freely, its velocity increases, and its pressure also varies during the interaction of gas–liquid two-phase flow. Figures 7–9 show the internal pressure distribution of the water drop at 0.02 s, 0.06 s and 0.10 s, respectively, and the white wire frame is the outer edge of the water drop. It can be seen from these figures that the internal pressures of water drops are high in the middle and relatively low around the center. At 0.02 s, the maximum internal central pressure is 31.73 Pa at 293.15 K, 31.50 Pa at 283.15 K, 32.72 Pa at 263.15 K, 32.36 Pa at 253.15 K and 32.52 Pa at 243.15 K. The pressure is the lowest at 283.15 K and the highest at 263.15 K, with a difference of 3.87%. At 0.06 s, the maximum internal central pressure is 32.91 Pa at 293.15 K, 32.03 Pa at 283.15 K, 35.45 Pa at 263.15 K, 33.78 Pa at 253.15 K and 33.31 Pa at 243.15 K. The pressure is the lowest at 283.15 K and the highest at 263.15 K, with a pressure difference of 10.68%.

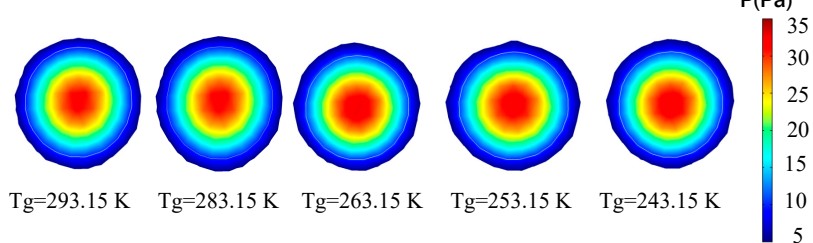

**Figure 7.** Pressure distribution cloud of the free-falling water droplet at t = 0.02 s. The white circle is the interface of water droplet and air.

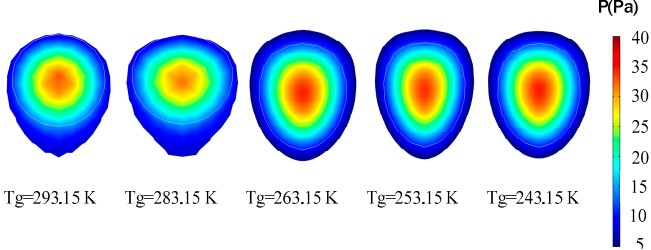

**Figure 8.** Pressure distribution cloud of the free-falling water droplet at t = 0.06 s. The white circle is the interface of water droplet and air.

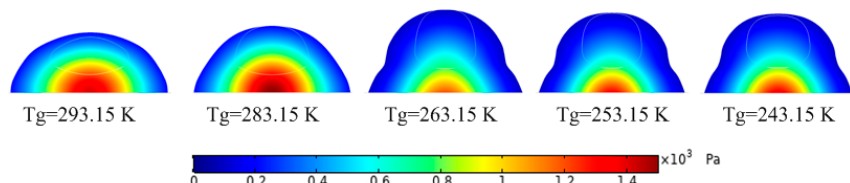

**Figure 9.** Pressure distribution cloud before water droplet impinging on bottom at t = 0.10 s. The white circle is the interface of water droplet and air.

Figure 9 shows the pressure distribution when the water droplet falls adjacent to the landing point. At 0.10 s, the pressure in the flow field increases markedly, and the highest pressure domain locates in front of the landing point. As shown in Figure 8, the highest pressure is 1407 Pa at 293.15 K, 1510 Pa at 283.15 K, 1203 Pa at 263.15 K, 1321 Pa at 253.15 K and 1342 Pa at 243.15 K. Because of the flat shape of the water droplet before landing, the pressure in freezing temperatures is lower than that in above zero temperatures. For example, the highest pressure is 1510 Pa at 283.15 K, but the lowest pressure is 1203 Pa at 263.15 K. The difference ratio is about 20.33%.

*6.3. Temperature Field*

In the drop of the two-phase water droplet considering temperature variation, there is heat transfer between the water droplet with a temperature of 283.15 K and the outside air. Except for the air at 283.15 K, the temperature distributions of water droplets and air in the other four cases are shown in Figures 10–12, and the black wire frame is the outer edge of the water droplets. In addition to the above velocity and deformation, the temperature distribution at 0.06 s and 0.1 s shows the heat exchange between the droplet and the surrounding air. The water droplets are heated gradually in air at 293.15 K, rising to 286 K at 0.06 s and about 289 K at 0.1 s. The flow field of the falling process shows that there is a low temperature region behind the droplet at 0.06 s and a low temperature wake at both sides of the droplet at 0.1 s. The water droplet is cooled in the frozen air.

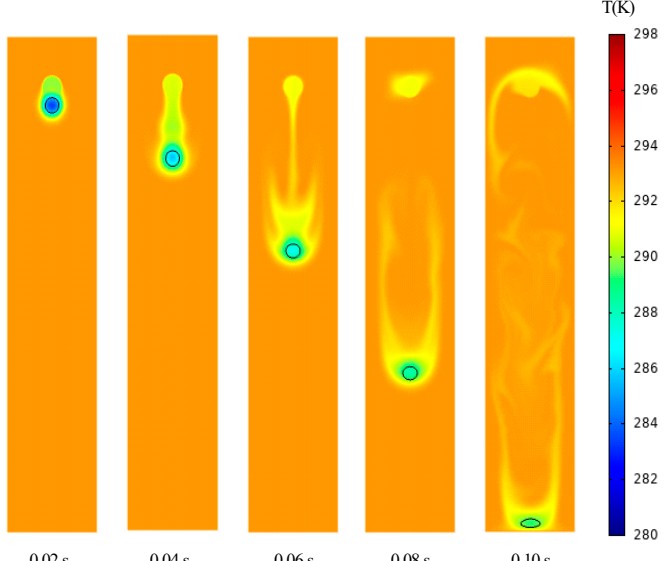

**Figure 10.** Temperature distribution of a free-falling water droplet in air at 293.15 K.

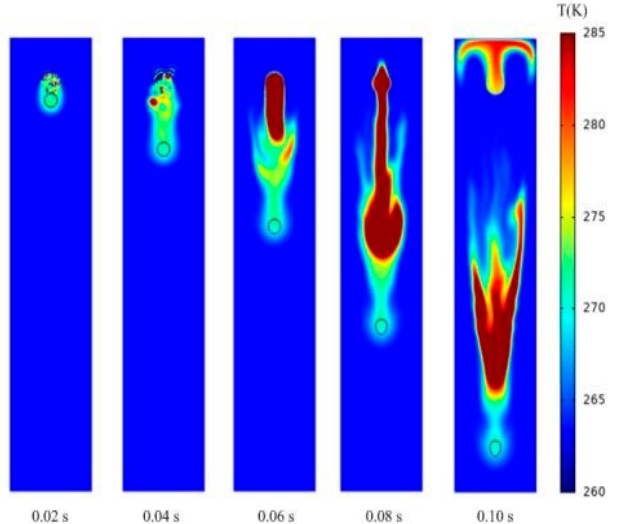

**Figure 11.** Temperature distribution of a free-falling water droplet in air at 263.15 K.

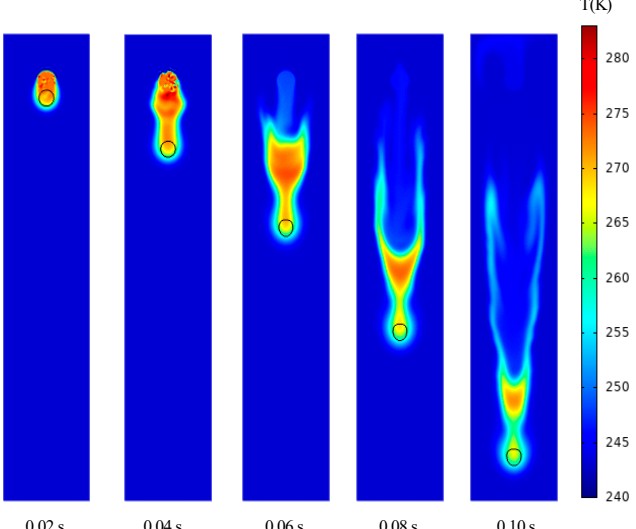

**Figure 12.** Temperature distribution of a free-falling water droplet in air at 243.15 K.

The temperature of water droplets and the surrounding air changes most acutely in the air at 263.15 K. at 0.06 s, and the temperature of water droplets drops to about 275 K. There is a cup-shaped high-temperature region at the tail of water droplets at about 280 K. At 0.1 s, the temperature of water droplets drops to about 273 K, and there is a necklace-shaped high-temperature region at the tail of water droplets. When the air temperature is as low as 253.15 K and 243.15 K, the droplet temperature decreases rapidly, and the change in ambient air temperature weakens. At 0.06 s, the internal temperature of the droplet drops to 275 K and 268 K, respectively, while at 0.1 s, the internal temperature of the droplet drops to 270 K and 260 K, respectively. The high-temperature region of the water droplet tail becomes weak and shallow.

During the water droplet falling in freezing temperatures, the viscosity of the water droplet varies with the decrease in temperature. Figures 13 and 14 show the viscosity of the water droplet falling in air at 263.15 K and 243.15 K, respectively. The water droplet is a whole round shape with a lower viscosity value at 0.02 s. Before landing on the bottom, the water droplet has a higher viscosity value of about 0.5 at 0.10 s. After landing on the bottom, the water droplet splits in two in the air at 263.15 K and 243.15 K. At 0.50 s, there are two of the same water droplets with semi-sphere shapes located on the bottom.

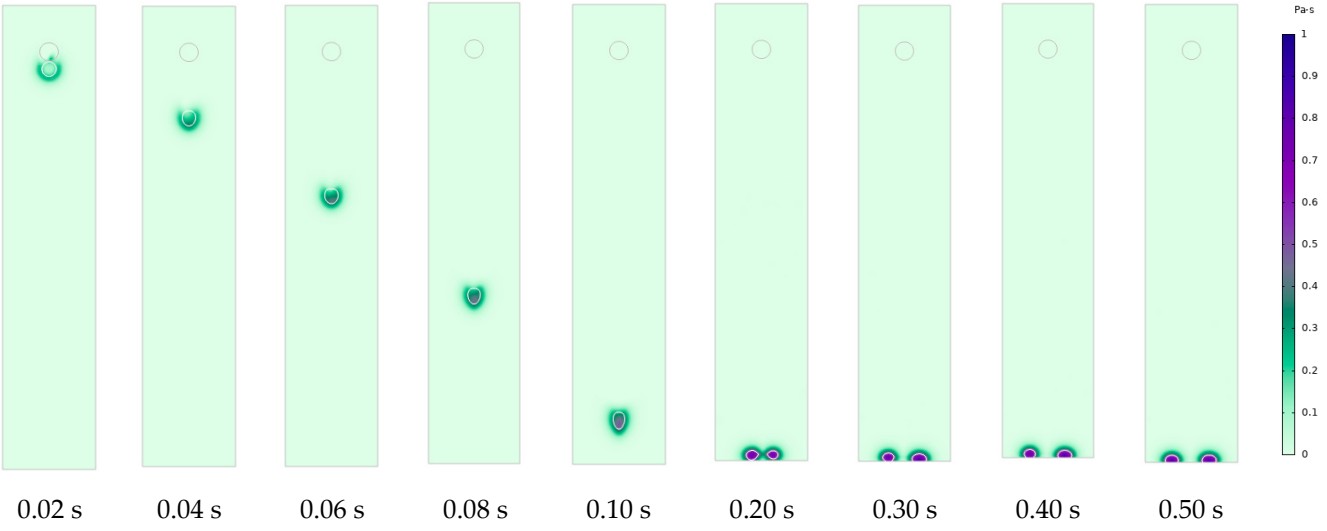

**Figure 13.** Viscosity of the water droplet in air at 263.15K.

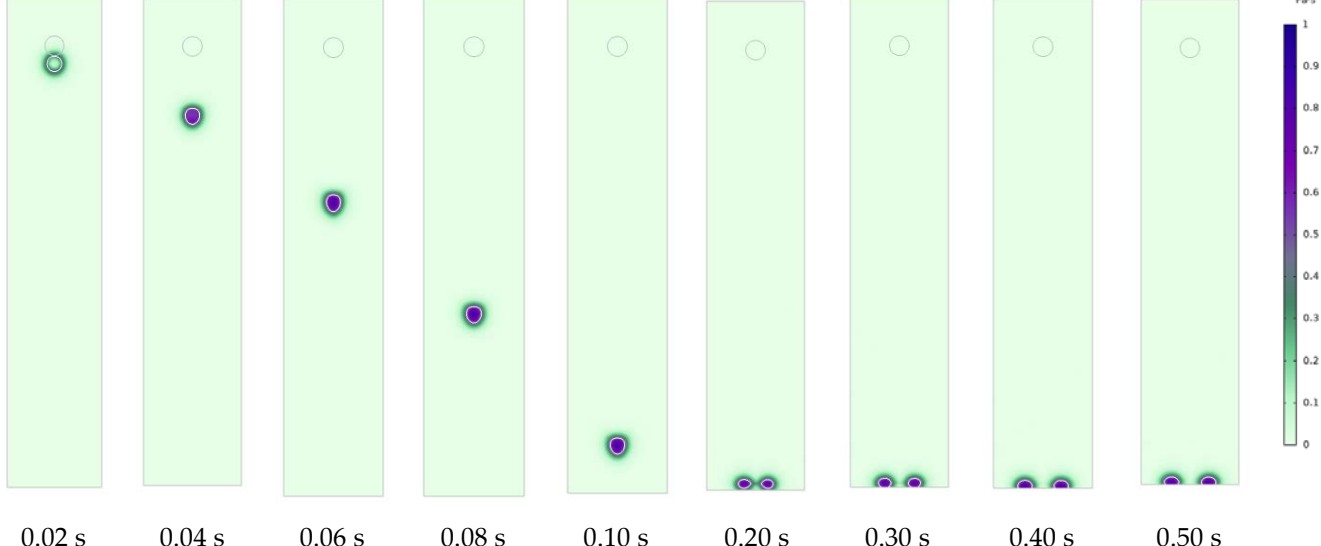

**Figure 14.** Viscosity of the water droplet in air at 243.15K.

### 6.4. Water Droplet Impacting on Bottom

During impacting on the bottom, the water droplet changes into different shapes at different temperatures. Figures 15–17 show the transient shape of the water droplet at 0.14 s, 0.20 s and 0.30 s, respectively. When the air temperature is 293.15 K, the water droplet splits into two domains at 0.14 s, and then they combine together at 0.20 s, but mostly move to the side at 0.30 s. When the air temperature is 283.15 K, the water droplet splashes out small droplets at 0.14 s, and then combines into two large water drops located on both sides of the mid small water droplet at 0.20 s, but changes into two mound-like-shaped water droplets at 0.30 s. The water droplet does not splash when the air temperature is 263.15 K, 253.15 K or 243.15 K. All the water droplets are squashed when they impact the bottom at 0.14 s, and then are divided into two small water droplets at 0.20 s and 0.30s, but there is a volume difference between the two water droplets. When the air temperature is as low as 243.15 K, the two water droplets almost have the same shape and the same volume at 0.30 s.

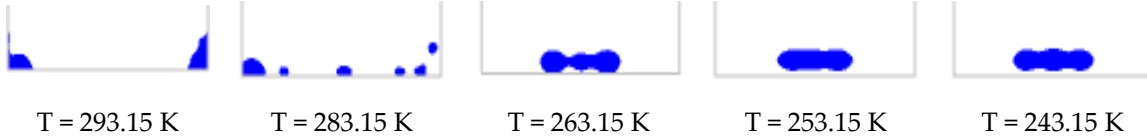

| T = 293.15 K | T = 283.15 K | T = 263.15 K | T = 253.15 K | T = 243.15 K |

**Figure 15.** Water droplets at 0.14 s. Grey line is the solid bottom; blue is water droplet.

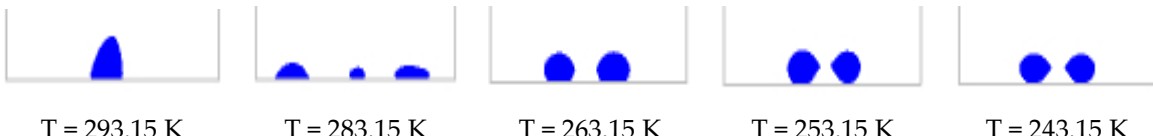

| T = 293.15 K | T = 283.15 K | T = 263.15 K | T = 253.15 K | T = 243.15 K |

**Figure 16.** Water droplets at 0.20 s. Grey line is the solid bottom; blue is water droplet.

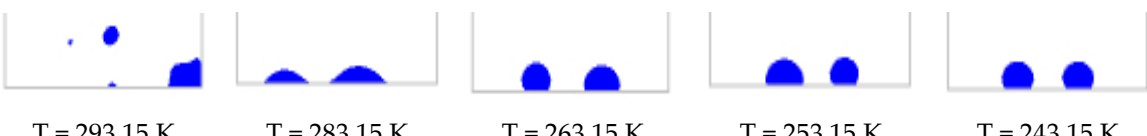

| T = 293.15 K | T = 283.15 K | T = 263.15 K | T = 253.15 K | T = 243.15 K |

**Figure 17.** Water droplets at 0.30 s. Grey line is the solid bottom; blue is water droplet.

### 6.5. Water Droplet Freezing on Bottom

After landing on the pipe bottom, the phase change of the attached water droplet occurred markedly when the air temperature was 263.15 K, 253.15 K and 243.15 K. The water droplet's temperature is 275.15 K, the contact angle is 90 degrees and the bottom has the same temperature as the air. The radius of the attached hemispherical water droplet is 2 mm, because the initial spherical water droplet of $R$ = 2 mm breaks up into two small water droplets after landing on the bottom. The coupling of layered flow and heat transfer is considered in the non-isothermal flow computation of the 2D model. Table 4 shows that the best accurate numerical computation is when the grids are in Refined 1. This grid type is used to calculate the freezing process.

**Table 4.** Independence of the freezing time of the water droplet at 243.15 K.

| Grids | Normal | Refined 1 | Refined 2 |
| --- | --- | --- | --- |
| Numbers | 1548 | 2948 | 6292 |
| Mean mesh quality | 0.8183 | 0.8527 | 0.8337 |
| Time(s) | 7.8 | 5.4 | 5.8 |

The three freezing processes are shown in Figures 18–20. When the attached water droplet stays in a static state, there is a quick phase change at 0.1 ms, and the phase change interface first occurs from the bottom. In Figure 18, when the ambient temperature is 263.15 K, the first phase change interface is likely a horizontal line before 10.0 ms, but the second phase change interface occurs from the top of the water droplet at 15.0 ms. The second phase change interface inserts in the water droplet straight, like a cylinder. After 15 ms, the second phase change interface spreads quickly, but the first phase change interface moves up slowly. At last, the two phase change interfaces combine. The total freezing time of the water droplet at 263.15 K is 11.2 ms. When the ambient temperature is 253.15 K and 243.15 K, the freezing processes are almost similar. The freezing time is 5.8 ms when the water droplet is in air at 253.15 K, and 3.8 ms in air at 243.15 K. Compared with the freezing in 263.15 K, the freezing time in air at 253.15 K and 243.15 K can be reduced by 30.77% and 74.04%. What is more, the phase change interface appears early from the top of the water droplet, and there is a visible interface that moves quickly from top to bottom during the freezing from 2.0 s to 3.5 s in air at 243.15 K.

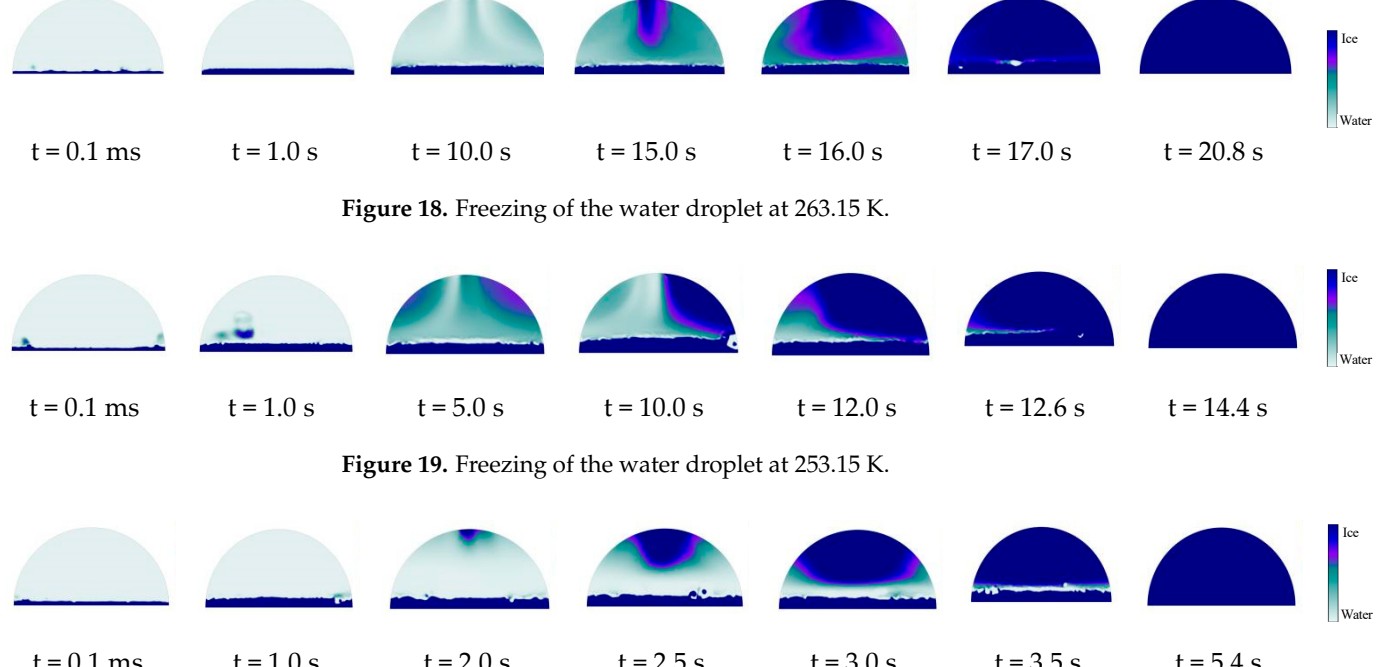

| t = 0.1 ms | t = 1.0 s | t = 10.0 s | t = 15.0 s | t = 16.0 s | t = 17.0 s | t = 20.8 s |

**Figure 18.** Freezing of the water droplet at 263.15 K.

| t = 0.1 ms | t = 1.0 s | t = 5.0 s | t = 10.0 s | t = 12.0 s | t = 12.6 s | t = 14.4 s |

**Figure 19.** Freezing of the water droplet at 253.15 K.

| t = 0.1 ms | t = 1.0 s | t = 2.0 s | t = 2.5 s | t = 3.0 s | t = 3.5 s | t = 5.4 s |

**Figure 20.** Freezing of the water droplet at 243.15 K.

## 7. Conclusions

In this paper, the hydrodynamic analysis of a water drop falling in air at mild temperature and sub-zero temperature is numerically simulated. Considering the two-phase flow, the shape, falling velocity, internal pressure and flow field temperature changes, the the free-falling water droplet in air at five specific temperatures is compared. This research and analysis are carried out combined with the characteristic parameters of hydrodynamics. The main conclusions are as follows:

(1) In the process of free falling, the water droplet deforms due to its surrounding forces. Before impinging on the bottom, the water droplet is the shape of a flat ellipse at 293.15 K, a drum at 283.15 K, a melon seed at 263.15 K and a human face at 253.15 K and 243.15 K when the falling time is 0.10 s.

(2) The falling velocity of the water droplet is affected by the temperature. In the process of falling in air at a sub-zero temperature, the water droplet experiences a continuous increase in its falling velocity. When the water droplet falls in air at a mild temperature, the falling velocity is slightly larger than that in air at a sub-zero temperature, and there is a velocity drop at 0.10 s due to the impinging on the bottom surface.

(3) There are great differences in the distribution of inner pressure and temperature. When the water droplet falls in air at a sub-zero temperature, there is a temperature increase behind the water droplet due to the latent heat released by ice freezing.

(4) In the air at a sub-zero temperature, the phase change of the water droplet first occurs at the bottom, and then propagates inwards from the top of the water droplet, and at last the phase change finishes at the inner of the water droplet with a line.

**Author Contributions:** Y.S.: conceptualization, methodology and writing—original draft preparation; H.G.: methodology and supervision; Y.Z.: software and data curation. All authors have read and agreed to the published version of the manuscript.

**Funding:** This research received no external funding.

**Data Availability Statement:** Not applicable.

**Acknowledgments:** This work has received help from the Institute of Refrigeration and Cryogenic Engineering and the Fundamental Research Funds for the Central Universities.

**Conflicts of Interest:** The authors declare no conflict of interest. The funders had no role in the design of the study; in the collection, analyses, or interpretation of data; in the writing of the manuscript; or in the decision to publish the results.

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
