# Peer review of "Numerical Analysis of the Free-Falling Process of a Water Droplet at Different Temperatures"

_processes, doi:10.3390/pr11010258_

Round 1
Reviewer 1 Report
Please see tha attached pdf file.

Author Response
Dear reviewer:
Thanks for your valuable suggestions.

Reviewer 2 Report
- There are only a few number of previous related research mentioned in the introduction. Please do a comprehensive literature review and bold the novelty of your work.
1. The first sentence of the abstract must be revised:... at SPECIFIC OPERATIONAL temperatures...
2. The whole text needs a deep modification in the English language. there are many errors related to s, the, .... For example:
...the "two-phase" flow...
... air and water dropletS...
....and icing of A water droplet...
These are only a few of the many mistakes in a few lines! Please check the whole text by a native English editor.
3. Heading numbers are wrong. 0-introduction!!! It seems that the authors have not checked the manuscript even once! Please check the numbers and orders everywhere.
4. Figure 4 needs a scale bar.
5. There are repeated numbers in Figures 14, and 15, ... Please check all the numbers
6. In figures 14-16 color bara are too small. They must be much bigger.
7. The authors must explain model validation and clarify the error of their simulations.
Author Response
Dear reviewer:
Thanks for your review.
